# The Combination of a Diversified Intake of Carbohydrates and Fats and Supplementation of Vitamin D in a Diet Does Not Affect the Levels of Hormones (Testosterone, Estradiol, and Cortisol) in Men Practicing Strength Training for the Duration of 12 Weeks

**DOI:** 10.3390/ijerph17218057

**Published:** 2020-11-01

**Authors:** Michał Wrzosek, Jakub Woźniak, Dariusz Włodarek

**Affiliations:** Department of Dietetics, Instutute of Human Nutrition Sciences, Warsaw University of Life Sciences (WULS—SGGW), Nowoursynowska 159 C, 02-776 Warsaw, Poland; kontakt@michalwrzosek.pl (M.W.); dariusz_wlodarek@sggw.edu.pl (D.W.)

**Keywords:** high-fat diet, high-carb diet, strength training, testosterone

## Abstract

This study aims to verify the extent to which a diversification of carbohydrates and fats intake in a diet, together with the reduction in vitamin D deficiency, impact the levels of hormones (testosterone, estradiol, cortisol) and Sex Hormone Binding Globulin (SHGB) in men doing strength training. The research involved 55 men aged 19–35. The participants were divided into two groups following two interventional diets for 12 weeks: high-fat diet (LCHF) or high-carb diet (LFHC), which were applied to satisfy the caloric requirements of each participant. Moreover, vitamin D supplementation was included. Moreover, both before and after following interventional diets, the level of hormones in participant’s blood was examined. After 12 weeks of following interventional diets in both groups, no changes of the levels of testosterone and estradiol, as well as SHGB, were found. The cortisol level in both groups decreased; however, only in the LFHC group was the change statistically significant (*p* = 0.03) and amounted to −3.5% (the cortisol level on an empty stomach in this group decreased from 14.17 ± 3.35 to 13.93 ± 2.63 mcg/dl). In both groups, the supplementation of vitamin D brought about a significant change in the level of vitamin D metabolite (25 (OH) D) (*p* = 0.01). In the LCHF group, the level of metabolite increased by 95%, and by 58.3% in the LFHC group. The increase in vitamin D metabolite in blood was higher in the LCHF group than the LFHC group, which might have resulted from the supplementation of this vitamin.

## 1. Introduction

Testosterone, the principal male sex hormone, impacts physical capacity and the increase in body mass. Massachusetts Male Aging Study (MMAS) research suggests that the level of testosterone in men has decreased in recent years [1]. The study involved blood sampling and collecting information about the health condition and biographical data from about 1500 randomly chosen men living in Boston, and was conducted for 17 years (1987–1989, 1995–1997, 2002–2004). The results showed that the testosterone levels in males tend to decrease over the decades. The cause of the decrease is unknown, and such factors as aging, obesity and smoking fail to account for these results. 

Both physical activity and a diet may influence the testosterone level in athletes. A well-balanced diet should accord with the physiology of the activity since the proper selection of nutrients helps maintain homeostasis in a body and proper levels of hormones [2] The current state of research shows the possible impact of diet composition on testosterone levels in men. Undoubtedly, the caloric value impacts the level of hormones in physically active people. Increased levels of energy intake, which lead to body fat build-up, may negatively influence testosterone levels [3]. However, the reduced levels of energy intake may also negatively impact testosterone levels, particularly in the long-term [4]. Training intensity and regeneration, which are crucial for optimal training adaptation, depend on a proper testosterone-cortisol ratio [2].

A low-carbohydrate diet may lead to an increase in cortisol levels and a decrease in free testosterone levels. Anderson et al. [5] studied the level of testosterone in seven men who followed a high-carbohydrate diet or high-protein diet for ten days. Their research showed that the level of testosterone in men following the high-carbohydrate diet was higher when compared to men following the high-protein diet. It follows from this that the proper carbohydrates to proteins ratio in a diet is an essential factor influencing the level of steroids in plasma [5]. Maintaining the proper carbohydrates levels is significant due to their impact on testosterone production. This results from two factors: the impact of insulin on 11β-hydroxysteroid dehydrogenase (including the reduction in cortisol production), and the impact of the level of glucose on the secretory capacity of gonadoliberin (GnRH) through the hypothalamus, which is directly related to a stimulation of pituitary to the production of luteinizing hormone (LH) which stimulates gonads to produce steroid hormones [6].

Furthermore, a possible impact of diets on the level of Sex Hormone Binding Globulin (SHGB) was noted. Longcope et al. [7] observed that a high-protein diet influenced the increase in the SHGB level. However, it has to be noted that they did not assess the protein intake in the diet before the intervention. Hence, it is possible that the increase in SHGB level resulted from the increase in protein intake, which was previously too low.

The impact of cholecalciferol intake on testosterone synthesis seems to be probable due to vitamin D receptors and enzymes metabolizing the vitamin in Leydig cells [8,9]. Observational studies showed the correlation between a low level of vitamin D metabolite (25 (OH) D) and a low level of androgens in participants. In one randomized research involving healthy men with vitamin D deficiency, the reduction in vitamin D deficiencies increased the level of total, free and bioactive testosterone [10,11,12].

The current research shows that low-calorie diets may negatively influence the level of testosterone and other hormones in physically active men. Moreover, the deficiency of vitamin D may negatively influence homeostasis. To determine the impact of cholecalciferol on the condition of male gonads, further research explaining the exact impact of vitamin D on testosterone production is necessary, especially, when taking into consideration fact that 89.9% of the Polish population shows too low concentration of the vitamin D metabolite (0.25 (OH) D), i.e., below 30 ng/ml of blood serum [13]. Moreover, the impact of macronutrient intake on homeostasis in men requires further examination. Hence, this study aims to examine how the diversity of fats and carbohydrates intake in a diet and the correction of vitamin D deficiency impact the levels of hormones (testosterone, estradiol, cortisol) and SHGB in men doing strength training.

## 2. Methodology

The present experiment involved 55 men aged 19–35 average 27.5 ± 3.37, with an average BMI of 24.01 ± 1.17 (min. 20.1, max. 26.1). The participants were required to: be male, have been doing strength training for at least six months, at least three times a week, prior to the experiment, demonstrate normal testosterone levels (300 ng/dl–800 ng/dl) and the lack of a chronic condition. The participants declared that they had conducted strength training continuously for the last 2–3 years. Moreover, they were amateurs practicing strength physical activity. The lowest training frequency was 3 training units during the week which usually lasted about 70 min. During the study, the participants did a training which included exercising all muscle groups during one training unit. The training itself was supervised by a qualified trainer who supervised the correctness of the exercises and encouraged the subjects to use training progression so that the training was effective. Training loads were individually selected according to the exercise capacity of the participants. The study included three stages.

The first stage consisted of the recruitment of participants by employing a Cavi method, assessment of the daily pattern of eating with the use of a 3-day food intake record [14], anthropometric measures—height, body mass, arms, waist and hips circumference, as well as the calculation of BMI (kg/m^2^). Moreover, the participants had their body composition examined using a bioelectrical impedance analysis (BIA) by employing Tanita MC-780 P analyzer. Finally, individual recommendations based on determined caloric requirements were arranged, which was to prepare the participant for the basic diet.

The second stage consisted of following the basic diet for two weeks. The diet was designed according to the recommendations for healthy adults [15]. The caloric value was in line with the energy requirements of each man participating in the research, which was based on a basic metabolic rate assessed with a bioelectrical impedance analysis including physical activity level [PAL], according to the Institute of Food and Nutrition [16]. The PAL of participants was between 1.5 and 1.8. The percentage of carbohydrates in the diet was set at 55% of the caloric value, including added sugars to 10%. The percentage of energy from fats constituted 25–35% of the caloric value of a diet, and the protein intake was set to 2 g per 1 kg of fat-free body mass, as specified in BIA [17]. The intake of vitamins and mineral components was calculated on the basis of norms for the Polish population [15]. The participants received a 7-day meal intake schedule that was individually customized to the preferences of taste, which was modified, albeit in adherence to the goals of the diet schedule, to keep the participants motivated to complete their tasks. Both diets were balanced with the use of a program including products and meals recommended by the Institute of Food and Nutrition and USDA.

After two weeks, we renewed the anthropometric measurements. Moreover, for the first time, we drew the participants’ blood to determine the level of total testosterone, estradiol (testosterone and estradiol were measured with the use of Roche test by the electrochemiluminescence method on Cobas system), SHGB, cortisol and 0.25 (OH) D. Blood was drawn in a diagnostic laboratory by a qualified nurse and according to legally binding standards.

In the third stage, two interventional diets were applied for 12 weeks, shortly after the completion of the second stage. The caloric value of the interventional diets remained the same as was individually determined in the second stage and was aligned with the participant’s energy requirements. The protein intake also remained unchanged and was set to 2 g per 1 kg of fat-free body mass. The participants were randomly assigned to two groups: the low-carb, high-fat diet (LCHF) (*n* = 27), and low-fat, high-carb diet (LFHC) (*n* = 28). In the LCHF diet, carbohydrates constituted up to 40% (mean 38.7 ± 6.3%) of the caloric value of the diet, and fats complemented the caloric deficiencies (mean 40.2 ± 8.8%) of the caloric value of the diet. In the LFHC diet, fats constituted 21.7 ± 1.9% of the caloric value of the diet, and carbohydrates complemented the caloric deficiencies (mean 58.2 ± 4.8%) of the caloric value of the diet.

All participants had a vitamin 0.25 (OH) D concentration below 30 ng/mL (mean 20.34 ± 9.28 ng/mL). During the experiment, all participants supplemented 2000 IU of vitamin D in the form of Vigantolleten, which has the status of a medicament according to the Polish standards. The composition of the general interventional diets is presented in Table 1, and Table 2 presents detailed characteristics.

The participants also received a 7-day meal intake schedule that was individually customized to the preferences of taste. During the research, the participants were regularly examined by the head of the experiment and submitted a report including their body mass and anthropometric measurements (arms, waist, hips and thigh circumference) every two weeks. At the beginning and end of the research, the measurements were conducted by the leader of the research. During the experiment, the anthropometric measurements were conducted by the participants, who had been trained to complete this task prior to the experiment. Moreover, they were asked to keep a food diary to monitor the real consumption of the diet. Each participant received strict guidelines regarding the completion of the diary. The notes included the list of all of the consumed products, meals and drinks, using household measures (e.g., cups, spoons) and/or mass units (grams), as well as the duration of their consumption. The food diary also included information about the types and daily amount of supplementation intake. In the event of a participant’s failure to adhere to the schedule, he was duly instructed to follow the guidelines. The proportions of ingredients were modified in adherence to the goals of the diet plan to keep the participants motivated to complete their tasks. At the end of the third stage, the results of anthropometric measurements and the examinations of the body composition through BIA were collected. Blood sampling was used to verify the levels of hormones (testosterone, estradiol), SHGB and 0.25 (OH) D. The scheme of the research is presented in Figure 1.

The quantitative study was conducted with the STATISTICA 13.3 PL software (TIBCO Software Inc. 2017). Moreover, basic descriptive statistics were conducted on the basis of two groups. Due to the rejection of normal distribution hypothesis of most analyzed variables by Shapiro–Wilk *W* test, nonparametric tests were applied in this research: the Mann–Whitney *U* test (with continuity correction), Wilcoxon matched-pairs test and Spearman’s rank correlation, respectively. To reject the null hypothesis, *p* < 0.05, the liminal value was applied throughout the testing procedure.

## 3. Results

Shortly before the implementation of the experimental diets, the median of participant’s body mass (*p* = 0.00001) differed and amounted to 83.4 kg in the LCHF group, and 76.6 kg in the LFHC group. The median of BMI in the LCHF group amounted to 24.7 kg/m^2^ and 23.9 kg/m^2^ in the LFHC group. The differences in BMI between the groups were statistically significant. The median of the absolute amount of body fat mass significantly differed between the groups. On the other hand, the participants’ body fat levels did not differ between the groups and amounted to 17.1% in the LCHF group, and 16.2% in LFHC group (Table 3).

Before the implementation of the experimental diets, no significant differences regarding the level of hormones (testosterone, estradiol, cortisol) and SHGB between the groups were found (Table 4). Moreover, the participants’ level of all hormones was within the limits of reference value for gender and age. In both groups, the deficit of vitamin D in all participants was found (lower than 30 ng/mL). The median of the level 0.25 (OH) D was 18.9 ng/mL in the LCHF group, and 22.2 ng/mL in the LFHC group.

This 12-week-long experiment, which was based on diets of diversified carbohydrates and fats intake, yielded the following results. In the LCHF group, no significant difference in the testosterone level was found (it increased only by 2.5%). In the LFHC group, the increase in testosterone level was higher and amounted to 12%. However, it continued to demonstrate no statistical significance. Moreover, the estradiol level in both groups did not significantly change. Although the cortisol level decreased in both groups, the change was statistically significant only in the LFHC group and amounted to −3.5% (in this group, the cortisol level on an empty stomach decreased from 14.17 ± 3.35 to 13.93 ± 2.63 mcg/dl) (Table 5). In both groups, a significant change in terms of the level of the vitamin D metabolite, 0.25 (OH) D, was found as a result of vitamin D supplementation. In the LCHF group, the level of the metabolite increased by 95%, and in the LFHC group by 58.3%. No relationship was observed between the increased concentration of vitamin D metabolite and the concentration of the tested hormones and SHGB (Spearman’s test, *p* > 0.05).

## 4. Discussion

The current state of research suggests that the level of testosterone in men has decreased in recent years [18]. However, we have noted that before and after our experiment, all men demonstrated a proper and physiological level of testosterone. It may indicate a positive impact of strength training. Moreover, it may be assumed that the amount and composition of fat in a diet may influence the levels of hormones in men. Some research confirms a negative impact of excessive fats intake in a diet on the male hormone [19,20]. In their research involving 12 athletes, Volek et al. [21] observed a significant correlation between monounsaturated fats intake and the testosterone level. Moreover, the experiments on rats [22,23] prove a positive correlation between unsaturated fats and the testosterone level. However, our research has shown that a both high-fat and high-carb diet does not significantly impact homeostasis in participants who at the early stage of the research demonstrated proper hormones levels, body mass, as well as were physically active. It is notable that although an increase in the testosterone levels was found in the participants after 12 weeks of following the diet, the result failed to demonstrate statistical significance. This result appears to confirm a low impact of diets of diversified carbohydrates and fats ratio on the secretion of male hormones. The fat intake in the LCHF group amounted to about 40% of the caloric value on average, whereas the LFHC group energy from fats intake amounted to about 22% of the caloric value of the diet. Moreover, the amount of particular fats did not have an impact on testosterone level in participants. Such significant differences in the composition of the diet did not significantly impact testosterone and estradiol levels in the examined men, whose levels of hormones were within the norm. Dorgan et al. [24] obtained different results in their study, which aimed to assess the impact of a diet on hormone levels in 43 men. The participants were divided into two groups. The members of the first group (A) were asked to follow a high-fat diet (about 40% of the caloric value) with a low amount of fiber. The second group (B) received a low-fat diet (about 20% of the caloric value) with a higher amount of fiber. The groups followed the diets for ten weeks. The result of the experiment showed that the daily testosterone production increased by 13% in the group following the high-fat diet. However, it has to be noted that the experiment was completed by men aged 19–56 who did not train regularly, and some of them were overweight, whereas our research was completed by physically active young male adults with a healthy body mass. Moreover, the average testosterone level in the examined group in the research of Dorgan et al. [24] was much lower than in ours. All of these factors might have impacted the obtained results. Hence, it may be assumed that the ratio of diversified carbohydrates and fats in a diet does not impact the testosterone level if physically active young men with a healthy mass maintain high protein intake and proper caloric value. This assumption is likely to be related to factors responsible for the synthesis of the hormone in a body.

The results of the morning testing sessions based on the participants of the examined group demonstrated a significant impact of carbohydrates and fats ratio on the level of cortisol in a diet. In the LFHC group, a significant decrease in its level was confirmed. This result may be related to the optimal re-synthesis of muscle glycogen in this group, which, in turn, translated into a faster post-workout regeneration. The pace of post-workout regeneration seems to have a significant impact on cortisol level in men, and the cortisol level proves an inverse correlation to testosterone level [25,26]. Such an increased impact of carbohydrates intake in a diet on cortisol level is also observed in other studies. Anderson et al. [26], in their experiment involving 20 men doing endurance sports, assessed the impact of a number of carbohydrates in a diet on the correlation between cortisol and testosterone. The participants were randomly assigned to a group with a low (30% of caloric value) or high (60% of caloric value) carbohydrates intake. The blood samples were taken at the early stage of the research, before and after each training, and 24 h after the last training. The participants completed a training involving a 60-min cycling session at 70–75% VO_2_ max for three consecutive days. A low-carbohydrate diet resulted in an increase in cortisol level and a decrease in free testosterone level. The proper proportion between cortisol and testosterone is the indicator of proper training intensity and regeneration, which is crucial for optimal training adaptation.

Vitamin D has a high impact on the human body. It is assumed that the expected concentration of vitamin D metabolite (25 (OH) D) in people practicing sports should be between 30 and 50 ng/mL [27]. It has been shown that the prevalence of vitamin D deficiency in society is very high [13]. The results of our experiments do not deviate from the trend as all participants had a 25 (OH) D concentration below the reference range of 30 ng/mL. Apart from its influence on the calcium-phosphate management of the body, it also impacts the immune system, hormone levels, including testosterone, and, notably, the athletes’ effort capacity [28,29]. The correlation between vitamin D and testosterone levels in men was assessed by Wehr et al. [10]. During a one-year-long experiment, they assessed the testosterone level and vitamin D metabolite (0.25(OH) D) in men’s serum. They observed a positive correlation of these parameters and seasonal variability of both parameters in men’s blood. The highest levels of 25 (OH) D and testosterone were recorded in the summer, and the lowest in the winter. UVB radiation stimulates the production of vitamin D in the skin, and the process is most intensive in the summer. However, there is no clear explanation for the fluctuation of testosterone level positively correlating with vitamin D levels. Some research suggests an indirect relation between vitamin D and testosterone and suggests that it results from the impact of cholecalciferol on inhibiting testosterone aromatization and the increase in the affinity of receptors binding androgens [8,30]. The increase in vitamin D levels may cause an increase in testosterone levels. However, this result has not been confirmed by a satisfactory number of studies [9,12], and, consequently, further research on this subject is required. This correlation might be also be caused by seasonal variations in the training routine. For instance, more high-intensity training sessions are likely to take place in the summer than in other seasons. In our research, the compensation of vitamin D deficit in both groups did not significantly increase the testosterone, estradiol and SHGB levels. Moreover, the impact of cortisol is inconclusive, since, in our experiment, the diet composition is more significant than the reduction in vitamin D deficiency. The results of our research show the lack of correlation between vitamin D levels in a body of physically active young men with a healthy body mass and the level of the hormones. Moreover, in another randomized and double-blind research involving 46 athletes, no impact of supplementation with 2000 IU of vitamin D on the testosterone level was observed [31]. It is possible that the impact of the supplementation of vitamin D on the level of hormones in athletes varies depending on a person and depends on the genetic variability of the protein binding vitamin D [32]. Another explanation may be higher enzyme activity in physically active persons, which, in turn, translates into higher demand for vitamin D in athletes [33].

## 5. Conclusions

To sum up, we did not observe significant differences in hormone levels resulting from the implementation of diets of diversified carbohydrates and fats intake. The only exception was the change of cortisol level. However, in spite of these variations, the concentration of cortisol was within acceptable reference norms. Although the level of the hormone in both groups decreased, the change was statistically significant only in the LFHC group. It probably resulted from the optimal re-synthesis of muscle glycogen in participants following the high-carbohydrate diet, which, in turn, translated into faster post-workout regeneration.

We have also noticed vitamin D deficiency in participants and a positive impact of vitamin D supplementation in quantities of 2000 IU. Moreover, the increase in vitamin D metabolites in participants’ bodies was higher in the high-fat diet group than in the high-carbohydrate diet group. This result may be related to the higher intake of this vitamin in a diet of higher fat intake, or better digestibility of this vitamin in the process of supplementation due to the higher lipid’s intake. At the same time, vitamin D supplementation enabled a significant increase in its concentration, regardless of the fat intake in the diet. No impact of the equalization of vitamin D deficiency on testosterone, estradiol, cortisol and SHGB levels has been shown.

## Figures and Tables

**Figure 1 ijerph-17-08057-f001:**
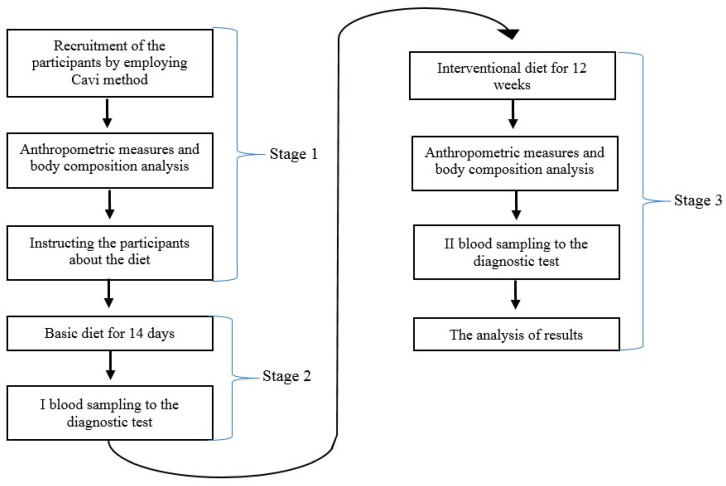
The scheme of the research. The local Ethics and Scientific Research on Humans Commission of Faculty of Human Nutrition and Consumer Sciences—SGGW (Warsaw University of Life Sciences) approved the research project (approval number: 17/2017).

**Table 1 ijerph-17-08057-t001:** General characteristic of interventional diets.

Variable	LCHF Group	LFHC Group
	(*n* = 27)	(*n* = 28)
	Mean ± SD	Mean ± SD
Caloric value [kcal]	The caloric value was in line with the energy requirements of each man participating in the research	
Proteins [g]	2.0 g/kg body mass (about 20% of energy)	
Fats in total [g]	40.2 ± 8.8%	21.7 ± 1.9%
Carbohydrates [g]	38.7 ± 6.3%	58.2 ± 4.8%

LCHF: Low Carb High Fat Diet; LFHC: Low Fat High Carb Diet.

**Table 2 ijerph-17-08057-t002:** Detailed characteristic of interventional diets.

Variable	LCHF Group	LFHC Group
	(*n* = 27)	(*n* = 28)
	Mean ± SD	Mean ± SD
Caloric value [kcal]	2938 ± 153.9	2687 ± 203.7
Caloric value [kcal/kg body mass]	35.36 ± 1.85	35.41 ± 2.68
Proteins [g]	142.5 ± 9.8	129 ± 16.2
Proteins [g/kg fat free body mass]	2.06 ± 0.13	2.01 ± 0.25
Fats in total [g]	131.3 ± 28.9	65 ± 5.8
Carbohydrates [g]	284.8 ± 46.9	391.2 32.9
Fiber [g]	41.7 ± 7	44.6 ± 6.9
Saturated fats [g]	33.4 ± 4.8	19 ± 2.9
Monosaturated fats [g]	67.9 ± 10.6	24.7 ± 3.5
Polysaturated fats [g]	25 ± 3.26	13.9 ± 2.6

**Table 3 ijerph-17-08057-t003:** Characteristics of groups at the early stage of the research.

Variable	LCHF Group		LFHC Group		*p* *
	(*n* = 27)		(*n* = 28)		
	Mean ± SD	Median	Mean	Median	
		(Min–Max)		(Min–Max)	
Age [years]	28.33 ± 3.01	29	26.71 ± 3.56	26.5	0.06
		(23–34)		(19–35)	
Height [cm]	184.16 ± 5.66	185	178.71 ± 5.07	178.3	0.0002
		(171–195)		(168–192)	
Body mass [kg]	83.07 ± 7.4	83.4	75.88 ± 4.79	76.6	0.00001
		(60.4–95.4)		(65–89)	
BMI [kg/m^2^]	24.34 ± 1.07	24.7	23.7 ± 1.2	23.9	0.03
		(20.4–26.1)		(20.1–25.6)	
Body fat mass [kg]	13.9 ± 3.21	14	11.76 ± 3.06	12.1	0.01
		(6.7–19.8)		(5.1–16.3)	
Fat content [%]	16.33 ± 3.24	17.1	16.98 ± 9.66	16.2	0.5
		(9.8–21.2)		(7.5–19.1)	
Free fat mass [kg]	69.17 ± 5.76	70.4 (53.7–78.9)	64.12 ± 3.87	64.2 (56.4–73.8)	0.0002
Body hydration [%]	59.46 ± 2.78	59.2	61.18 ± 3.30	60.9	0.07
		(55–64.7)		(55.9–69.9)	

* Mann–Whitney *U* test.

**Table 4 ijerph-17-08057-t004:** The mean and median of the levels of hormones and Sex Hormone Binding Globulin (SHGB) in both groups at the early stage of the research.

Variable (Reference Range)	LCHF Group		LFHC Group		*p* *
	(*n* = 27)		(*n* = 28)		
	Mean ± SD	Median	Mean ± SD	Median	
		(Min–Max)		(Min–Max)	
Testosterone [ng/dl]	569.44 ± 134.85	542	589.39 ± 152.14	613	0.64
(249.00–836.00)		(319–830)		(249–825)	
Estradiol [pg/mL]	25.81 ± 8.27	24.4	25.03 ± 7.88	23.9	0.71
(11.30–43.20)		(14.1–43)		(11.1–40.7)	
Cortisol [mcg/dl]	14.17 ± 3.04	14.6	14.17 ± 3.35	15	0.56
(4.82–19.5)		(7.7–19.3)		(8.6–19.8)	
SHGB [nmol/l]	42.21 ± 16.19	39.8	44.46 ± 15.15	41.6	0.4
(18.30–54.10)		(21.5–52)		(21.5–50.9)	
0.25 (OH) D [ng/mL]	20.34 ± 9.28	18.9	23.52 ± 10.52	22.2	0.18
(30.00–50.00)		(8–29)		(9.9–29.1)	

* Mann–Whitney *U* test.

**Table 5 ijerph-17-08057-t005:** Mean and median of the levels of hormones and SHGB after the intervention, and the changes in the levels of hormones and SHGB in both groups.

Variable	LCHF Group				LFHC Group			
	(*n* = 27)				(*n* = 28)			
	Mean ± SD	Median	Change	*p* *	Mean ± SD	Median	Change	*p* *
		(Min–Max)				(Min–Max)		
Testosterone [ng/dl]	577.18	592	2.50%	0.75	625.57	591	11.90%	0.2
	±144.79	(281–810)			±138.1	(376–881)		
Estradiol [pg/mL]	23.87	23.2	−3.4%	0.08	23.72	21.6	−2%	0.24
	±8.79	(12–41)			±8.23	(11–40.8)		
Cortisol [mcg/dl]	13.16	13	−4.5%	0.09	13.93	14.2	−3.5%	0.03
	±2.75	(8–19.9)			±2.63	(8.5–18.4)		
SHGB [nmol/l]	42.65	40.1	2.90%	0.77	46.25	43.4	8.80%	0.87
	±16.14	(22–51)			±17.39	(21.9–53.1)		
0.25 (OH) D [ng/mL]	34.25	31	95%	0.01	32.79	33.5	58.30%	0.01
	±8.91	(21–56)			±7.46	(18.9–49.9)		

* Wilcoxon matched-pairs test.

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
