# Peer review of "The Combination of a Diversified Intake of Carbohydrates and Fats and Supplementation of Vitamin D in a Diet Does Not Affect the Levels of Hormones (Testosterone, Estradiol, and Cortisol) in Men Practicing Strength Training for the Duration of 12 Weeks"

_ijerph, 2020, doi:10.3390/ijerph17218057_

Round 1

Reviewer 1 Report

Dear Authors,

Your study looked like a failed experiment since no statistically significant change was found in the key elements of your study. Your response confirmed my concerns: Your aim was not to test if an intervention (diet) made significant changes in a dependent variable (level of hormones) but to verify the extent to which a diversification of fats and carbohydrates intake in a diet as well as compensation for the possible. An experimental study should test if a hypothesis is accepted or not, so "to verify the extent to which" seems a fuzzy statement, not adequate to a scientific study. Your results don't go either against or in favor of the current literature because you cannot come to any conclusion of the type: when y increases, we see an increase in x (or the contrary).

My suggestion is to repeat the experiment with more than 12 weeks to allow for proper statistical changes between participants. 

Author Response

We highly appreciate reviewers' comments and we try to take them into account. However, we do not understand the above comments. A successful experiment does not have to end up showing significant differences. The lack of differences is also the result. Striving to obtain significantly different results at all costs proves the lack of objectivity and reliability in the approach to their interpretation. Our research allows us to verify the influence of various diets on the tested parameters and shows that there is no such influence.

Should we keep researching until the results are significantly different? The question arises whether the year would be sufficient and whether, if no differences were observed at that time, the study should be continued to force the parameters to change.

The research time as well as the entire methodology were consulted with the scientific community (scientific meetings) and approved by the ethics committee, which also assessed the methodical side of the work.

Human research involving active participation of the examined group is extremely difficult. We managed to engage as many as 55 people to follow the complicated research procedure.

We believe that our study provides significant data on the relationship between the composition of the diet and the hormonal balance of physically active men.

Reviewer 2 Report

Article title: The combination of a diversified intake of carbohydrates and fats and supplementation of vitamin D in a diet does not affect the levels of hormones (testosterone, estradiol, and cortisol) in men practicing strength training for the duration of 12 weeks.

Article number: ID: ijerph-945322

Authors presented the evaluation of hormones level in relation to the type of diet: Low Fat High Carb and High Fat Low Carb.

The study has been design and performed correctly. There are some limitation (the cortisol was detected at single time points which give small diagnostic value, however that was for study purposes and in whole participants the measurements were done at the same time, of the highest value during the day/night cycle).

The statistical analysis are performed correctly and results presented in a clear and understandable way. The tables are clear and give the overview of the obtained results.

The obtained results give no differences between diet and the level of hormones except the cortisol that was higher in LFHC group (statistically significant).

There are two questions that the information is missing in the manuscript:

  1. If the first blood collection and measurements were done after 14 weeks of basic diet, how the participants were classified (it was mentioned that they should have a normal testosterone level). Was this measurements done before participants classification or after the 14 days basic diet?
  2. It would be interesting from the dietary point of view how the BMI, body fat mass and fat content changed during 12 weeks of diet supplementation?

Author Response

Point 1: If the first blood collection and measurements were done after 14 weeks of basic diet, how the participants were classified (it was mentioned that they should have a normal testosterone level). Was this measurements done before participants classification or after the 14 days basic diet?

Response 1:

All the hormones mentioned in the article were tested after the 14-day basic diet. In the event of incorrect test results (excluding the participant from the experiment), he would have been informed about it, however, after the first stage of the study, the results of hormone tests did not reveal their too high level in any case.

Point 2: It would be interesting from the dietary point of view how the BMI, body fat mass and fat content changed during 12 weeks of diet supplementation?

Response 2:

It is a very interesting issue, however, due to the abundance of these data, we have decided to describe them in another article in which we focus on the impact of the diversified fats and carbohydrates intake on the body composition of men who do strength sports.

Point 3: line 114: typo low-fat, high-fat diet (LFHC) ==> low-fat, high carb diet (LFHC).

Response 3:

We appreciate you have noticed this error. We have corrected it.

Reviewer 3 Report

The current study, titled " The combination of a diversified intake of carbohydrates and fats and supplementation of vitamin D in a diet does not affect the levels of hormones (testosterone, estradiol, and cortisol) in men practicing strength training for the duration of 12 weeks" attempted to verify the extent to which a diversification of carbohydrates and fats intake in a diet, together with the reduction of vitamin D deficiency in a study group of 55 men for the duration of 12 weeks period.

Major Points

  1. The authors divided study subject two group, LCHF (~40% fats, ~40% Carbs) and LFHC (~20% fats, ~60% Carbs). The difference in caloric values between these two groups. LFHC diet, designed as ~20% fats and ~60% Carbs is more like everyday normal diet rather than interventional diet: the second stage diet, so called the basic diet for two weeks, consisted of ~55% Carb + 25~35% fats. Authors need to provide the rationale how they decided composition of the interventional diet. Personally, I don't think ~20% changes in the composition of diet would have any significant impact on body metabolism.

  1. Authors need to explain why the anthropometric values (age, height, body mass, body fat) between two experimental groups are significantly different and why they did not group recruited subjects with anthropometric values equally distributed.

Minor

line 114: typo low-fat, high-fat diet (LFHC) ==> low-fat, high carb diet (LFHC).

Author Response

Point 1: The authors divided study subject two group, LCHF (~40% fats, ~40% Carbs) and LFHC (~20% fats, ~60% Carbs). The difference in caloric values between these two groups. LFHC diet, designed as ~20% fats and ~60% Carbs is more like everyday normal diet rather than interventional diet: the second stage diet, so called the basic diet for two weeks, consisted of ~55% Carb + 25~35% fats. Authors need to provide the rationale how they decided composition of the interventional diet. Personally, I don't think ~20% changes in the composition of diet would have any significant impact on body metabolism.

Response 1:

The methodology of selecting the composition of the diet was based on the assumption that in the first two weeks the macronutrients intake would be similar in both groups. Then, when establishing the composition of the interventional diets, we applied the limit values for the fat intake consistent with the principles of proper nutrition. Indeed, the 20% difference in the proportion of fat in the diet doubles the supply of fat in a high-fat diet compared to a low-fat diet. It is a very significant change in the intake of fatty acids

as well as carbohydrates while maintaining the energy value of the diet in line with the participants' needs. The 20% fat content corresponds to the low-fat diets used in the studies of other authors, and the 40% corresponds to the high-fat diets. Also, the selection of the limit values ​​of the fat content resulted from the requirements set by the ethical committee, whose experts agreed to such a maximum diversification of fat intake bearing in mind the participants’ safety. The differences in the energy value of the diet resulted from the differences in body weight and, therefore, in the total energy needs of men classified into particular groups. The assumption was that the energy needs of the participants were fully met. At the same time, the energy value of the diet per kg of body weight did not differ between the groups, which eliminated the influence of differences in the energy value of the diet on the obtained results and allowed to assess the influence of only the composition of the diet.

Point 2: Authors need to explain why the anthropometric values (age, height, body mass, body fat) between two experimental groups are significantly different and why they did not group recruited subjects with anthropometric values equally distributed.

Response 2:

Human studies are very complex, and the participants need to clearly declare their willingness to participate in the study and to follow their diet during a 12-week experimental diet. Therefore, in order to divide the participants into two groups, we used simple randomization with the selection of people for a given group, taking into account their nutritional preferences disclosed in the completed questionnaire at the beginning of the study, so that the participants of the study were more willing to continue its assumptions. Due to the voluntary participation of the groups, we found that simple randomization is better than matching which would be impossible to do correctly in the studied group of people.

Point 3: line 114: typo low-fat, high-fat diet (LFHC) ==> low-fat, high carb diet (LFHC).

Response 3:

We are grateful for noticing the mistake. We have corrected it.

Reviewer 4 Report

The article intituled "The combination of a diversified intake of carbohydrates and fats and supplementation of vitamin D in a diet does not affect the levels of hormones (testosterone, estradiol, and cortisol) in men practicing strength training for the duration of 12 weeks." reports the article analyzed the effects of exercises, diet and Vitamin D in steroid hormone levels. 

There are some general questions:

1- Why the authors did not use a control group without exercises with the two diets? At least references about the effect of these diets without exercise. Also, it is important to correlate exercises and Vitamin D levels in the introduction

Specific comments:

Introduction:

Add references about exercises and Vitamin D levels

Is there any reference about non-exercise people and vitamin D

Also, it could be interesting to add some references about the levels of Vitamin D in the general population establishing that all participants were into the normal range.

Methodology:

Why 55 participants?

Why 12 weeks?

Does the diet contain cholesterol?

I believe that is important to describe in more detail what kind of exercises were done. Are the participant's athletes?

It also could be interesting to have, cholesterol, blood glucose and insulin levels of the participants. Since the average BMI in the LCHF group is in the limit. Even though body mass fat is normal.

The difference between Vitamin D in the High-fat diet may be related to this high fat.

in the results tables, it would be interesting to add the normal expected range in the population

The author comment that they observed a deficiency in the levels of vitamin D, but 20 ng/mL, but that are several ranges in literature and is important to add the normal ranges in literature with references.

Even though the results did not demonstrate differences between the groups, the conclusion could be better explored.  

Author Response

Point 1: Why the authors did not use a control group without exercises with the two diets? At least references about the effect of these diets without exercise. Also, it is important to correlate exercises and Vitamin D levels in the introduction

Response 1:

The assumption of the study was to check how dietary intervention affects the parameters in exercising people. Undoubtedly, it is interesting to see how it affects people who are not physically active, however, it was not the aim of our experiment. In the available literature, we have not found information that physical activity alone affects the concentration of vitamin D. Of course, lifestyle and the frequency of exposure to solar radiation may be an influencing factor, nevertheless, it is an indirect influence. Therefore, we have not described this issue.

Point 2: Introduction:

Add references about exercises and Vitamin D levels

Is there any reference about non-exercise people and vitamin D

Also, it could be interesting to add some references about the levels of Vitamin D in the general population establishing that all participants were into the normal range.

Response 2:

We have added references regarding the level of vitamin D deficiency in Polish society.

Point 3: Methodology:

Response 3:

Why 55 participants?

Such a number of people agreed to participate in an experiment with a complicated procedure. At the same time, this number enables statistical analyzes to be made. Statistical methods were selected to be adequate for small groups.

Why 12 weeks?

This time was due to the need to obtain a stable concentration of vitamin D, and the occurrence of possible changes in hormonal tests. It was also important to obtain the participants’ consent for such a long observance of the dietary regime and research procedures. Also, the ethics committee agreed to the maximum 12 weeks, considering the time to be sufficient.

Does the diet contain cholesterol?

The diets were balanced in terms of the content of dietary macronutrients, other than the presented ones, including cholesterol, in accordance with the nutritional recommendations.

Point 4: I believe that is important to describe in more detail what kind of exercises were done. Are the participant's athletes?

Response 4:

The participants declared that they had done strength training continuously for the last 2-3 years. Moreover, they were amateurs practicing strength physical activity. The lowest training frequency was 3 training units during the week which usually lasted about 70 minutes. During the study, the participants did a training which included exercising all muscle groups during one training unit. The training itself was supervised by a qualified trainer who supervised the correctness of the exercises and encouraged the subjects to use training progression so that the training was effective. Training loads were individually selected according to the exercise capacity of the participants. We have included this information.

Point 5: It also could be interesting to have, cholesterol, blood glucose and insulin levels of the participants. Since the average BMI in the LCHF group is in the limit. Even though body mass fat is normal.

Response 5:

To take part in the experiment, the participants could not suffer from any chronic diseases and metabolic disorders. Therefore, after we chose the participants, we did not additionally evaluate these parameters. If we understand the remark correctly, the examined groups had different BMI, indeed, however, the difference resulted from simple randomization, which we considered the best method of dividing the participants into two groups. Nevertheless, the relative share of body fat in the two groups did not differ.

Point 6: The difference between Vitamin D in the High-fat diet may be related to this high fat.

in the results tables, it would be interesting to add the normal expected range in the population

Response 6:

We agree with this remark. We have described the issue in the discussion of the article.

Point 7: The author comment that they observed a deficiency in the levels of vitamin D, but 20 ng/mL, but that are several ranges in literature and is important to add the normal ranges in literature with references.

Even though the results did not demonstrate differences between the groups, the conclusion could be better explored.

Response 7:

We have added recommendations and reference values, as well as we have elaborated our conclusions.

Round 2

Reviewer 1 Report

Dear Authors,

I have read your rebuttal with great interest and I agree with you that a successful experiment does not have to end up showing significant differences. However, the problem with this manuscript is that you stated that "Undoubtedly, the caloric value impacts the level of hormones in physically active people", and that "The current research shows that low-calorie diets may negatively influence the level of testosterone and other hormones in physically active men", so there is a clear link between caloric diets and level of hormones. Then, your aim is to go a little further by studying the macronutrient intake in active men that may explain the relationships already mentioned. Your results lead to no statistical conclusion, even though you recruited 55 subjects during 12 weeks. Therefore, my suggestion still stands.

Author Response

Thank you for your reply. The low energy value of a diet may affect the hormones levels, therefore, in our study, we made sure that the energy value of the diet met the energy needs of the participants. Throughout the study, participants were given a diet consistent with their needs in terms of energy, protein and micronutrients. In this way, it became possible that the only differentiating factor was the proportion of fat and carbohydrates.

It has been described in the aims and methodology section. The subject of our study was not to determine the impact of energy deficiency or excess on hormone levels. We have used an isocaloric diet. Therefore, we cannot respond to the comments sent, which, in our opinion, are not relevant to our work.

Reviewer 3 Report

Authors responded appropriately to the comments. 

Author Response

Thank you for your review.

Reviewer 4 Report

After the revision and considerations made by the authors, I believe that the article is acceptable for publication. The only suggestion is that the normal levels of the hormones in the population should be added in table four, even though the author did comment that they are in the normal range.

Line 201- please rephrase

Author Response

Thank you for your review.

We added the normal levels of the hormones in the population  in the Table 4 and we rephrased line 201.

This manuscript is a resubmission of an earlier submission. The following is a list of the peer review reports and author responses from that submission.

Round 1

Reviewer 1 Report

Dear Authors,

The main problem of your manuscript is all about three hormones (testosterone, estradiol, cortisol) and Sex Hormone Binding Globulin, which were supposed to change after intervention. Results showed no change in any of them for the two groups under test. Even though the groups received different diet, there were no statistical differences in any hormone. Therefore, it seems a failed experiment, at least from the point of view that is raised. Your entire speech is focused on changes as well as your discussion. Even the title is confusing because it indicates an impact of the diet on the level of hormones, when there is no impact. Any reader viewing this work will think that this is other evidence for change, not a wayward article refuting the current trend.

Besides, the manuscript is in a developing stage:

  • Title wrong (perhaps two titles written together)
  • Abstract wrong: The first sentence is missing (no aim or description of the experiment)
  • Tables wrongly formatted (there is even one with all vertical and horizontal lines present)
  • Absence of Conclussion sectin

Other changes:

Line 20: Define SHGB: Sex Hormone Binding Globulin (also in Introduction)

Line 81: provide SD of age

Line 91 and others: please, change decimal comma to decimal point

Line 125: tables wrongly formatted

Line 131: Who did the anthropometric measurements every two weeks?

Line 166: In the Results section, the authors keep repeating the information of Tables in the text. Please, choose if you want to give that information on tables or on text, but not both.

Line 176: What is the meaning of scope?

Line 186: tables wrongly formatted

Author Response

Response to Reviewer 1
Point 1: The main problem of your manuscript is all about three hormones (testosterone,
estradiol, cortisol) and Sex Hormone Binding Globulin, which were supposed to change after
intervention. Results showed no change in any of them for the two groups under test. Even
though the groups received different diet, there were no statistical differences in any
hormone. Therefore, it seems a failed experiment, at least from the point of view that is
raised. Your entire speech is focused on changes as well as your discussion. Even the title is
confusing because it indicates an impact of the diet on the level of hormones, when there is
no impact. Any reader viewing this work will think that this is other evidence for change, not
a wayward article refuting the current trend.
Response 1: The aim of this study was to verify the extent to which a diversification of
fats and carbohydrates intake in a diet as well as compensation for the possible
deficiency of vitamin D influence the level of hormones (testosterone, estradiol, and
cortisol) and SHGB in men practising strength training. As the influence of the
modification of intake of these macro-ingredients as well as the compensation for
vitamin D deficiency on the levels of hormones has not been confirmed, we have
specified the title to avoid confusion.
The updated version of the title: The combination of a diversified intake of carbohydrates
and fats and supplementation of vitamin D in a diet does not affect the levels of hormones
(testosterone, estradiol, and cortisol) in men practicing strength training for the duration of 12
weeks.
Point 2:
 Title wrong (perhaps two titles written together)
 Abstract wrong: The first sentence is missing (no aim or description of the
experiment)
 Tables wrongly formatted (there is even one with all vertical and horizontal lines
present)
 Absence of Conclussion sectin
Response 2: The title has been modified to reflect the purpose and meaning of the research.
We have included the goals of the research at the beginning of the abstract. We have
formatted the table following the reviewer’s suggestions. We have added the conclusions
section.
Point 3:
Line 20: Define SHGB: Sex Hormone Binding Globulin (also in Introduction)
Line 81: provide SD of age
Line 91 and others: please, change decimal comma to decimal point
Line 125: tables wrongly formatted
Line 131: Who did the anthropometric measurements every two weeks?
Line 166: In the Results section, the authors keep repeating the information of Tables in the
text. Please, choose if you want to give that information on tables or on text, but not both.
Line 176: What is the meaning of scope?
Line 186: tables wrongly formatted
Response 3: We have explained the term SHGB. We have included standard deviation
including age average. We have changed commas into full stops in numerical values. We
have formatted the table. At the beginning and end of the research the measurements
were conducted by the leader of the research during the sessions. During the
experiment, the anthropometric measurements were conducted by the participants, who
were trained in this respect prior to the experiment. We have included this information
in the text. We have considered the suggestions concerning the presentation of the
results. The word scope refers to the lowest and highest obtained value in a given
parameter.

Reviewer 2 Report

The research involved 55 men aged 19-35. The participants were required to have been doing strength sports for at least six months, at least three times a week, before the experiment. The participants were divided into two groups following two interventional diets for 12 weeks: high-fat diet (LCHF) or high-carb diet (LFHC), covering the calorific requirements of each participant. Also, vitamin D supplementation was included. Moreover, both before and after following interventional diets, the level of hormones in the participant’s blood was examined. After 12 weeks of following interventional diets in both groups, no changes in the levels of testosterone and estradiol, as well as SHGB, were found. The cortisol level in both groups decreased; however, only in the LFHC group the change was statistically significant (p=0,03) and amounted to -3,5% (the cortisol level on an empty stomach in this group decreased from 14,17 ± 3,35 to 13,93 ± 2,63 mcg/dl). In both groups, the supplementation of vitamin D brought about a significant change in the level of vitamin D metabolite (25 (OH) D) (p=0,01). In the LCHF group, the level of the metabolite increased by 95%, and by 58,3% in the LFHC group. The exception was the decrease in cortisol levels in the LFHC group. The increase of vitamin D metabolites in blood was higher in the LCHF group than the LFHC group, which might have resulted from a better digestibility of this vitamin from supplementation on a high-fat diet.

in General: it's a good paper

Title: the title properly explain the purpose and objective of the article>

Abstract: abstract contains an appropriate summary for the article, language used in the abstract easy to read and understand, there are no suggestions for improvement.

Introduction: authors do provide adequate background on the topic and reason for this article and describe what the authors hoped to achieve.

Results: the results presented in a clear manner, the authors provide accurate research results, there is sufficient evidence for each result.

Conclusion: in general: Good and the research provides ample data for the authors to make their conclusion.

Grammar: Need Some revision.

Author Response

Response to Reviewer 2

Point 1:  Grammar: Needs Some revision.

Response 1: We appreciate your feedback regarding the language. We have made required changes in the article.

Reviewer 3 Report

There are a lot of issues with English to improve, for example:

The name of the article - The effect of high fat and high carbohydrate diets and vitamin D supplement on the level of hormones in males doing strength exercises

13 to have been doing -> to do

14 strength sports -> strength training - everywhere in the article

16 Calorific -> caloric - everywhere in the article

83 proper -> normal

92 consisted in -> consisted of everywhere in the article

95 metabolism ->metabolic

116 percentage of fat and carbohydrate in both diets should be specified

206 the number of fats in a diet - what does it mean? kinds if fats? amount of fat?

218 the number of particular fats -> the types of fats?

228 do sports -> train

253 Delete "as a fat-soluble vitamin

End so on and so forth

The results are not very impressive, maybe because 12 weeks are too short a period.

Even if the cortisol level in the high carb group decreased, and it is statistically significant, the change is still within the normal level and has no clinical significance.

It is expected that after Vitamin D supplementation, its level will go up, and that in the high fat group the level of 25(OH)D3 will  increase, because the vitamin is fat soluble.

Author Response

Response to Reviewer 2
Point 1: There are a lot of issues with English to improve, for example:
Response 1: We appreciate your feedback regarding the language. We have made required
changes in the article.
Point 2: The name of the article - The effect of high fat and high carbohydrate diets and
vitamin D supplement on the level of hormones in males doing strength exercises
Response 2: The aim of this study was to verify the extent to which a diversification of fats
and carbohydrates intake in a diet as well as compensation for the possible deficiency of
vitamin D influence the level of hormones (testosterone, estradiol, and cortisol) and SHGB in
men practising strength training. As the influence of the modification of intake of these
macro-ingredients as well as the compensation for vitamin D deficiency on the levels of
hormones has not been confirmed, we have specified the title to avoid confusion.
The updated version of the title: The combination of a diversified intake of carbohydrates
and fats and supplementation of vitamin D in a diet does not affect the levels of hormones
(testosterone, estradiol, and cortisol) in men practicing strength training for the duration of 12
weeks.
Point 3: percentage of fat and carbohydrate in both diets should be specified
Response 3: We have included the exact percentage of fat and carbohydrates in the text
and the table.
Point 4: 206 the number of fats in a diet - what does it mean? kinds if fats? amount of fat?
218 the number of particular fats -> the types of fats?
Response 4: We meant the amount of fat in a diet. We have clarified the issue in the
text.
Point 5: The results are not very impressive, maybe because 12 weeks are too short a period.
Even if the cortisol level in the high carb group decreased, and it is statistically significant,
the change is still within the normal level and has no clinical significance.
It is expected that after Vitamin D supplementation, its level will go up, and that in the high
fat group the level of 25(OH)D3 will increase, because the vitamin is fat soluble.
Response 5: Indeed, the period of the research might have been too short to observe
hormonal changes in participants. However, the fact that after 12 weeks no changes of the
levels of hormones were noted seems interesting. We have included information in the article
that although the cortisol levels in participants changed, they were within the norm. In
conclusions, we have provided the possible causes of the change of vitamin D levels in
participants.

Round 2

Reviewer 1 Report

Dear Authors,

Thank you for your detailed response to my suggestions. As I stated in my first review, your study looked like a failed experiment, since no statistical significant change was found in the key elements of your study. Your response confirmed my concerns: Your aim was not to test if an intervention (diet) made significant changes in a dependent variable (level of hormones) but  to verify the extent to which a diversification of fats and carbohydrates intake in a diet as well as compensation for the possible. An experimental study should test if an hyphotesis is accepted or not, so "to verify the extent to which" seems a fuzzy statement, not adequate to a scientific study. Your results don't go neither against or in favour of the current literature because you cannot come to any conclusion of the type: when y increases, we see an increase in x (or the contrary).

My suggestion is to repeat the experiment with more than 12 weeks to allow for proper statistical changes between participants. 

Reviewer 3 Report

The text was improved. Still the work is avarage.